# Rapid On-Site Identification for Three Arcidae Species (*Anadara kagoshimensis*, *Tegillarca granosa*, and *Anadara broughtonii*) Using Ultrafast PCR Combined with Direct DNA Extraction

**DOI:** 10.3390/foods11162449

**Published:** 2022-08-14

**Authors:** Ga-Young Lee, Eiseul Kim, Seung-Min Yang, Hae-Yeong Kim

**Affiliations:** Institute of Life Sciences & Resources, Department of Food Science and Biotechnology, Kyung Hee University, Yongin 17104, Korea

**Keywords:** *Anadara kagoshimensis*, *Tegillarca granosa*, *Anadara broughtonii*, on-site identification, ark shell adulteration, direct ultrafast PCR

## Abstract

Granular ark (*Tegillarca granosa*), broughton’s ribbed ark (*Anadara broughtonii*), and half-crenate ark (*Anadara kagoshimensis*) are important fishery resources throughout Asia; granular ark exhibiting a higher economic value due to its rarity. However, due to the similar morphological characteristics of the three species, the less valuable species could be exploited for food fraud. In this study, we developed a rapid on-site identification method based on a microfluidic chip for the detection of the three ark shell species. We designed new species-specific primers, targeting the genes encoding mitochondrial cytochrome b or cytochrome c oxidase I, for the identification of the three ark shells and estimated their specificity against 17 species, which amplified only the target species. The sensitivity of each primer was 0.001 ng. In addition, this method was further improved to develop a direct ultrafast polymerase chain reaction (PCR) for on-site food monitoring, which would allow for completing the entire procedure (from sampling to obtaining the results) within 25 min without DNA extraction. Our direct, ultrafast PCR was successfully applied to differentiate the three species from 29 commercial products. Therefore, this assay could be used as a rapid and cost-effective approach for the on-site identification of ark shells in commercial food products.

## 1. Introduction

Ark shells belong to the phylum Mollusca, class Bivalvia, order Arcida, and family *Arcidae*. They are commonly found on the western Pacific coast and inhabit muddy sediments of the shallow coasts at depths ranging between 10 and 150 m [1,2]. Ark shells are commercially important seafood in Asian countries, especially in Korea, China, and Japan, where they have been cultivated for a long time due to their good taste and high nutritional value [3]. In Korea, granular ark (*Tegillarca granosa*), broughton’s ribbed ark (*Anadara broughtonii*), and half-crenate ark (*Anadara kagoshimensis*) are commercially important ark shell species for fishery resources [4]. Due to their economic and taxonomic importance, ark shells have been the subjects of research efforts [5,6].

The three above-mentioned ark shell species can be morphologically distinguished, e.g., based on the fluff on the surface of the shell and the number of radial ribs. However, ark shell meats display very similar morphological characteristics; the morphological identification of the species is thus difficult when the edible parts are separated from the shell in the flesh form during processing [7]. In general, fresh ark shells are consumed as processed food, steamed, boiled, or seasoned to extend the shelf-life and ensure safety. Therefore, it is impossible to distinguish the three ark shell species from commercially processed products based solely on their morphological characteristics, potentially leading to an increase in economically motivated adulteration or false declarations of the raw material used [8]. Therefore, in order to detect food-related fraud and protect consumer interest, ark shell species authentication methods in various processed products should be developed.

Polymerase chain reaction (PCR) techniques, which can detect trace amounts of DNA with high specificity and sensitivity, are widely used to authenticate raw materials in food [9,10]. PCR-based techniques, such as DNA barcoding, PCR restriction fragment length polymorphism, forensically informative nucleotide sequencing, and recombinase polymerase amplification, have been developed to detect different species [11,12,13,14]. These techniques have been used to identify commercially important seafood species, such as shrimps, cods, crabs, and oysters [8,15,16,17]. Recently, more rapid and convenient technologies have been developed to confirm the presence of target species in processed foods [18,19]. Among several technologies, ultrafast PCR assay using microfluidic chips allows PCR reaction completion within 20 min and target species identification via melting curve analysis [20]. Moreover, this technique enables real-time visualization of the results without subsequent electrophoresis using label-free intercalating EvaGreen dye [20]. In fact, ultrafast PCR systems have been used to identify closely related eukaryotic species such as shrimp allergen, meat species, and mi-iuy croaker [18,21,22].

Despite their increasing value as food resources and potential spoilage, efficient and rapid identification assays have not yet been developed to discriminate the above-described three closely related ark shell species to prevent economically motivated food adulteration. The aim of this study is to develop a rapid and cost-effective ultrafast PCR assay combined with direct DNA extraction for field identification and the monitoring of the three ark shell species in processed products.

## 2. Materials and Methods

### 2.1. Samples

The reference specimen samples of granular ark (*Tegillarca granosa*), broughton’s ribbed ark (*Anadara broughtonii*), and half-crenate ark (*Anadara kagoshimensis*) were obtained from the National Institute of Biological Resource (NIBR, Incheon, Korea). The morphological characteristics of granular ark, broughton’s ribbed ark, and half-crenate ark were investigated based on specimen information provided by the National Fisheries Research & Development Institute in Korea (https://www.nifs.go.kr/frcenter/species/, accessed on 1 July 2022). The non-target species, including Venus mactra (*Mactra quadrangularis*), Japanese littleneck (*Venerupis philippinarum*), hard-shelled mussel (*Mytilus coruscus*), Japanese abalone (*Haliotis discus hannai*), horned turban (*Turbo cornutus*), Ocean quahog (*Arctica islandica*), lamellated oyster (*Ostrea denselamellosa*), Kuruma prawn (*Marsupenaeus japonicus*), Whiteleg shrimp (*Litopenaeus vannamei*), fleshy prawn (*Fenneropenaeus chinensis*), gazami crab (*Portunus trituberculatus*), American lobster (*Homarus americanus*), Japanese common squid (*Todarodes pacificus*), and Pacific chub mackerel (*Scomber japonicus*), were purchased in local markets in Korea. A total of 29 commercial ark shell products were purchased in markets in Korea. All samples were washed and immediately stored at −20 °C until further analysis.

### 2.2. DNA Extraction

DNA extraction of the samples was performed using the DNeasy Blood & Tissue Kit (Qiagen, Hilden, Germany), according to the manufacturer’s instructions. Briefly, 25 mg of raw and processed sample was lysed, and genomic DNA was extracted using all buffers specified in the manufacturer’s protocol. A spectrophotometer (Maestrogen, Las Vegas, NV, USA) was used to measure the A260/A280 absorbance ratio, determining DNA purity. DNA with A260/A280 values between 1.8 and 2.0 was used in this study. The extracted DNA was stored in a −20 °C freezer until further use.

### 2.3. Primer Design

The sequences of the mitochondrial Cytb and COI genes of 17 different shellfishes or seafood species were downloaded from the National Center for Biotechnology Information (NCBI, http://www.ncbi.nlm.nih.gov, accessed on 15 July 2022) and aligned by the Clustal Omega alignment system [23]. Three species-specific primer sets targeting specific regions were designed using the Primer Design program version 3.0 (Scientific and Educational Software, Durham, NC, USA) and synthesized by Bionics (Seoul, Korea). Table 1 summarizes all primer-related information.

### 2.4. In Silico Specificity Test

The specificity of newly designed primer sets was evaluated by web-based in silico PCR amplification [24]. The sequences of each primer set were aligned against ten and 32 mitochondrial genomes for target and non-target species, respectively, obtained from the GenBank database: *T*. *granosa* (NC_026081.1, KJ607173.1, and MW222474.1), *A*. *broughtonii* (OM807134.1, OM807133.1, OM807132.1, and OM807131.1), *A*. *kagoshimensis* (OM807136.1, MN366013.1, and KF750628.1), *A. antiquata* (MK783262.1), *A. consociata* (MH535977.1), *A. cornea* (OM807135.1), *A. crebricostata* (MN316632.1), *A. globosa* (MN366011.1), *A. gubernaculum* (MN061840.1), *A*. *inaequivalvis* (MN366012.1), *A. pilula* (KU975162.1), *A. transversa* (MN326817.1), *A. vellicata* (KP954700.1), *Arca zebra* (MN366003.1), *Arctica islandica* (KF363952.1 and KF363951.1), *Atrina pectinata* (KC153059.1), *Barbatia decussata* (MW629559.1), *Corbicula fluminea* (MK392334.1), *Crassostrea gigas* (MZ497416.1), *Crassostrea nippona* (HM015198.1), *Dreissena polymorpha* (CM035931.1), *Haliotis discus hannai* (KU310896.1, KF724723.1, and KU310897.1), *Mactra quadrangularis* (MN317133.1 and MW691169.1), *Meretrix lusoria* (MT418596.1), *Mizuhopecten yessoensis* (FJ595959.1), *Mytilus coruscus* (KJ577549.1), *Mytilus edulis* (AY484747.1), *Solen strictus* (NC_017616.1), *Turbo cornutus* (MZ826276.1), *Venerupis philippinarum* (AB065375.1 and AB065374.1).

### 2.5. Ultrafast PCR Assay Specificity and Sensitivity

Primer set specificity was tested using DNA extracted from 17 shellfish or seafood species. The standard curve was generated to determine ultrafast PCR efficiency and sensitivity. The standard curve was constructed by using serial dilutions of the target DNA from 10 ng to 0.0001 ng. The different DNA concentrations were plotted against the corresponding cycle threshold (Ct) values. The amplification efficiency was calculated based on the formula: Efficiency = 10^1/slope^ − 1 [25].

Ultrafast PCR was performed using a GenChecker UF-150 Real-time PCR system (Genesystem, Daejeon, Korea) with a microfluidic PCR chip, Rapi:Chip (Genesystem), a rapid chip with good thermal conductivity. The PCR reactions were set in 10 μL mixtures containing 5 μL of SSoFast EvaGreen SuperMix (Bio-Rad, Berkeley, CA, USA), 1 μL of the primer sets, and 10 ng of the template DNA. The ultrafast PCR reaction was performed under the following conditions: pre-denaturation for 1 min at 95 °C, followed by 40 cycles of denaturation at 95 °C for 5 s, annealing at 60 °C for 5 s, and extension at 72 °C for 5 s. The melting curve was generated at 60–90 °C to estimate the melting temperature of the PCR product. The amplified product was visualized through electrophoresis to confirm ultrafast PCR sensitivity. Electrophoresis was performed at 150 V for 15 min on a 2% agarose gel stained with ethidium bromide.

### 2.6. Direct Ultrafast PCR Assay Application on Processed Foods

An ultrafast PCR assay with direct DNA extraction was used to assess the on-site applicability of the ultrafast PCR. A total of 29 commercial ark shell products were used to verify the reliability of the direct ultrafast PCR assay. Briefly, 30 mg of commercial ark shell product was put into a tube, and 200 μL of 1× direct lysis buffer (Rapi:Driect^TM,^ Genesystem) was added, followed by mixing at room temperature for 3 min. After centrifugation at 13,600× *g* for 1 min, the supernatant was diluted 1/5 times with Tris-EDTA (TE, 10 mM Tris-HCl + 1 mM EDTA, pH 8.0) buffer, and 2 μL of the diluted sample was used as a template in the direct ultrafast PCR reaction. The direct ultrafast PCR conditions were the same as those of the ultrafast PCR described in Section 2.4. The accuracy of the direct ultrafast PCR assay was evaluated by comparing it with the ultrafast PCR reaction using the DNA extracted according to Section 2.2.

## 3. Results and Discussion

### 3.1. Morphological Characteristics

Appendix A presents the morphological characteristics of granular ark (*Tegillarca granosa*), broughton’s ribbed ark (*Anadara broughtonii*), and half-crenate ark (*Anadara kagoshimensis*). The three species showed morphological differences such as the fluff on the surface of the shell and the number of radial ribs. Granular ark exhibited 17 to 18 radial, evenly spaced ribs on the shells, being the smallest compared to those of other species. Broughton’s ribbed ark and half-crenate ark displayed 39–44 and 30–34 radial ribs, respectively, and their shell surfaces were covered with a brown fluffy shell. The adult size of the species in descending order was as follows: Broughton’s ribbed ark (approximately 7 cm), half-crenate ark (approximately 4 cm), and granular ark (approximately 3 cm). However, the appearance of the ark shell meats was very similar, i.e., almost indistinguishable in processed food products by appearance inspection.

### 3.2. Primer Design

The mitochondrial genome is well established to provide variable regions useful for eukaryotic identification and to study evolutionary relationships between species [26]. The genetic variation in mitochondrial genome sequences, such as single nucleotide polymorphisms and insertion and deletion variants (indels), have been useful tools for phylogenetic and population genetic analysis in eukaryotes [27]. Furthermore, the mitochondrial gene can sensitively detect a target even when the DNA in the samples is degraded or mixed with a small amount of DNA in food due to the high copy number of DNA in the cell [28]. Therefore, the sequences of different mitochondrial genome regions, such as 18S ribosomal RNA, 16S rRNA, Cytb, COI, and D-loop region, have been successfully used for authentication studies [8,20,29,30,31,32]. However, the efficiency of using mitochondrial genomes for differentiation among different ark shell species has not yet been investigated. For the first time, we have developed mitochondrial-based PCR markers to identify the three aforementioned ark shell species.

In this study, we used complete mitochondrial genome sequences of 17 different shellfishes and other seafood including target (granular ark, broughton’s ribbed ark, and half-crenate ark) and non-target (venus mactra, Japanese littleneck, hard-shelled mussel, Japanese abalone, horned turban, ocean quahog, lamellated oyster, kuruma prawn, whiteleg shrimp, fleshy prawn, gazami crab, American lobster, Japanese common squid, and Pacific chub mackerel) species to identify variable regions suitable for developing an ultrafast PCR assay. We aligned 17 genome sequences to design new primer sets that could specifically amplify only the target species (Figure 1). Specific primer sets for broughton’s ribbed ark and half-crenate ark were designed in the variable region of the mitochondrial COI region (accession numbers: AB729113.1 and KF750628.1, respectively) and the mitochondrial Cytb region (accession number: MW222474.1) for granular ark (Table 1).

### 3.3. In Silico Specificity Test

In silico, PCR is an efficient method to ensure primer specificity in a wide range of PCR applications, such as molecular diagnosis, microorganism detection, and forensic DNA typing [33]. The web-based in silico PCR amplification tool (http://insilico.ehu.es/PCR/, accessed on 15 July 2022) allows confirming the specificity of the designed primer set by combining local and global alignment algorithms [34].

Primer set specificities were confirmed using in silico analysis against primer binding sites on mitochondrial genomes. The primer sets targeting mitochondrial regions showed a high specificity, demonstrated by sequence similarity analysis with 42 mitochondrial genomes and no cross-reactivity with sequences from non-target species (Table 2). The amplicon sizes ranged between 105 and 112 bp. Primers for detecting thermally processed food, in which DNA can be degraded, should target multicopy genes and short sequences, thereby increasing the possibility of DNA amplification and improving the sensitivity compared to low-copy DNA targets [35]. The newly designed primer sets target mitochondrial DNA, a multicopy gene, and display a small amplicon size, suggesting it is suitable for identifying granular ark, broughton’s ribbed ark, and half-crenate ark and processed food containing them.

### 3.4. Ultrafast PCR Assay Specificity and Sensitivity

The specificity of the three primer sets was evaluated using 17 different shellfish and other seafood species. Each primer set only amplified the DNA of each target species, with no false positives among the non-target species (Table 3). The amplified product Ct values from each target species were 20.41 ± 0.08, 19.97 ± 0.05, and 21.53 ± 0.32 for granular ark, broughton’s ribbed ark, and half-crenate ark, respectively. The analysis of the melting curve allows calculating the melting temperature and confirms the absence of non-specific amplification-related unwanted DNA fragments [35]. The melting temperatures of the target species-related amplicons were 75.96 ± 0.41 °C, 76.25 ± 0.19 °C, and 77.72 ± 0.41 °C for granular ark, broughton’s ribbed ark, and half-crenate ark, respectively. Even if the melting temperatures of the three ark shell species were similar, target primer set-dependent specific analysis of the species would still be possible in each hole of the ultrafast PCR machine [20].

The standard curve should have slope values ranging between −3.1 and −3.6, corresponding to amplification efficiencies within the range of 90 to 110% to achieve a satisfactory efficiency level of the PCR-based method [36]. Standard curves for granular ark, broughton’s ribbed ark, and half-crenate ark were constructed using serially diluted genomic DNA of each species (10 ng to 0.0001 ng) independently and in triplicates to confirm the accuracy of the ultrafast PCR method. The slope values for granular ark, broughton’s ribbed ark, and half-crenate ark were −3.496, −3.573, and −3.516, respectively (Figure 2). The standard curves revealed efficiencies of 93.23%, 90.49%, and 92.49% for granular ark, broughton’s ribbed ark, and half-crenate ark. These results suggest that the three primer sets could efficiently identify the three species using ultrafast PCR.

Ultrafast PCR detectability was determined when three positive signals were confirmed in the PCR reactions, where less than two positive signals were considered negative results (Table 4). The sensitivity of the ultrafast PCR assay for granular ark, broughton’s ribbed ark, and half-crenate ark was 1 pg for each genomic DNA sample (Table 4). The ultrafast PCR amplicons were confirmed by gel electrophoresis (Appendix A). Previous studies reported detection limits for certain seafood: Kang et al. (2019) developed a species-specific PCR method for snow crabs (*Chionoecetes opilio* and *C*. *japonicus*) with a detection limit of 0.005 ng [8]. Wilwet et al. (2021) described a detection limit of 0.1 ng for seven commercial shrimp species (*Penaeus monodon, P. vannamei, P. semisulcatus, Fenneropenaeus indicus, Metapenaeus affinis, M. rosenbergii,* and *H. gibbosus*) [15]. Compared with previous studies, ultrafast PCR yielded a similar or two orders of magnitude higher sensitivity than that of conventional PCR methods. The lower detection limit of the PCR method makes it advantageous for processed seafood products [15]. Therefore, the suggested ultrafast PCR method could be used as a sensitive method to identify the three ark shell species as well as enable rapid detection using an ultrafast PCR system with the microfluidic chip.

### 3.5. Direct Ultrafast PCR Assay Application on Processed Food Products

To develop an on-site method for the rapid identification of the three ark shells in processed food products, a direct amplification method with no DNA extraction was combined with the ultrafast PCR system. To evaluate its applicability in the field, the direct ultrafast PCR assay was applied to 29 commercial food products, classified into nine product types: raw, boiled, seasoned, pickled, canned, porridge, auto, dried, and fried (Table 5). The direct ultrafast PCR assay accuracy was verified by comparing the DNA extracts of each food sample.

The analysis showed that 25 products contained the species declared by the producer (Table 5). Granular ark (*Tegillarca granosa*), broughton’s ribbed ark (*Anadara broughtonii*), and half-crenate ark (*Anadara kagoshimensis*) were detected in seven, eight, and ten products, respectively. The four products that did not indicate the accurate species name on the label contained half-crenate ark (*Anadara kagoshimensis*). The ultrafast and direct ultrafast PCR results were consistent with each other, suggesting that the direct ultrafast PCR method is very accurate in identifying the three ark shells from processed products. Processed foods with spice products reportedly contain inhibitors, such as polysaccharides, carbohydrates, lipids, and proteins, that can negatively affect the PCR [37,38]. PCR inhibitors contained in processed foods may affect PCR efficiency. However, in this study, ultrafast and direct ultrafast PCR efficiencies were the same, indicating that the PCR inhibitor in processed food did not affect the direct ultrafast PCR efficiency.

To the best of our knowledge, this is the first study that reported the development of a rapid and reliable ultrafast PCR assay using a microfluidic chip to identify three ark shell species. In this study, highly sensitive, mitochondrial gene-targeting primer sets were validated using ultrafast PCR technology, and they were further improved to develop an ultrafast PCR assay combined with direct DNA extraction. The developed ultrafast PCR assay does not require expensive equipment, such as a real-time PCR machine, and direct DNA extraction enables time- and cost-efficient analysis. The combination of these advantages allowed for the development of a method by which the entire procedure of sampling, DNA extraction, amplification, and data confirmation could be completed within approximately 25 min. The time required to detect seafood species in processed foods by using this method is less than that using other PCR-based methods [8,15,39,40]. Due to the advantages of quick analysis, ultrafast PCR or fast real-time PCR assays have been used to develop methods for the identification of economically motivated food adulteration [9]. Combining direct DNA extraction and ultrafast PCR allowed us to develop a rapid, cost-effective, and handy monitoring method for on-site analysis. Therefore, the ultrafast PCR assay combined with direct DNA extraction provides an approach for accurate on-site identification of three ark shell species from processed products.

## 4. Conclusions

In this study, granular ark-, broughton’s ribbed ark-, and half-crenate ark-specific primer sets were designed and used to develop an ultrafast PCR assay for the identification of the three species. The specific primer sets were designed based on the genes encoding mitochondrial cytochrome b (Cytb) or cytochrome c oxidase I (COI) to distinguish granular ark, broughton’s ribbed ark, and half-crenate ark. We used the designed primer sets to develop the identification method using an ultrafast PCR machine with a microfluidic chip. The ultrafast PCR assay combined with direct DNA extraction allowed for the successful, simple, and reliable authenticity verification of 29 commercial ark shell products. The direct ultrafast PCR assay could be performed without any DNA extraction, thereby enabling time-efficient analysis. In addition, this assay allowed for the completion of the entire analytical procedure within 25 min, enabling efficient on-site commercial food monitoring. Our assay could be efficiently used in various fields, such as rapid monitoring, quality control, and food fraud regulation, by the food industry and regulatory authorities. The assay developed in this study can be utilized for quantitative analysis using real-time PCR or droplet digital PCR equipment in the future.

## Figures and Tables

**Figure 1 foods-11-02449-f001:**
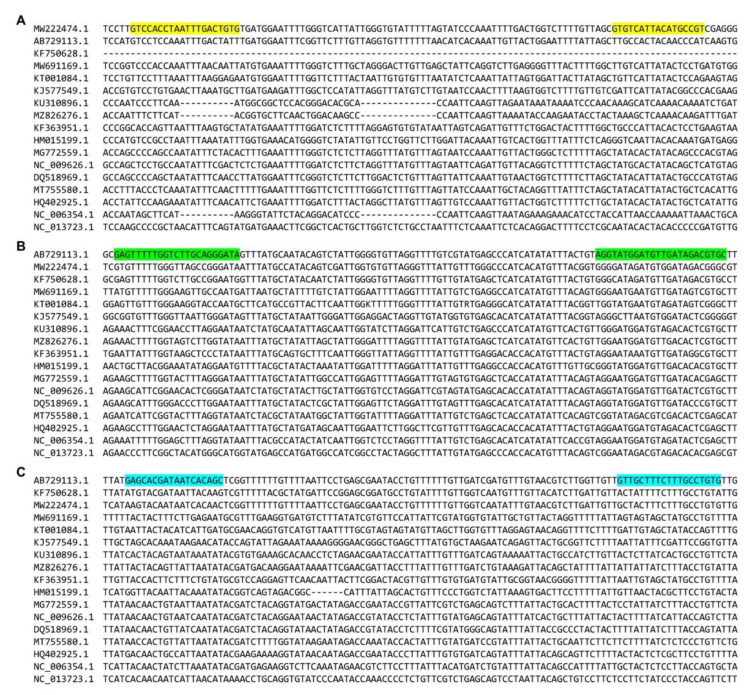
Sequence alignment of the mitochondrial genomes. (**A**) Alignment of cytochrome b (cytb) region against 17 different shellfishes or seafood species and *T*. *granosa*-specific primer site (yellow sequences). (**B**) Alignment of cytochrome oxidase subunit I (COI) region against 17 different shellfishes or seafood species and *A*. *broughtonii*-specific primer site (green sequences). (**C**) Alignment of COI region against 17 different shellfishes or seafood species and *A*. *kagoshimensis*-specific primer region (blue sequences).

**Figure 2 foods-11-02449-f002:**
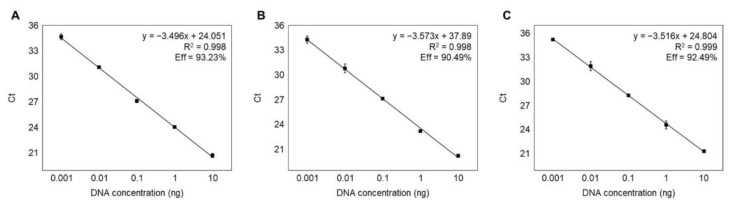
Standard curves by Ct values against DNA concentrations for (**A**) granular ark, (**B**) broughton’s ribbed ark, and (**C**) half-crenate ark. All samples were tested in triplicate.

**Table 1 foods-11-02449-t001:** The information on primer sets developed in this study.

Target Species	Name	Sequences (5′ → 3′)	Target	Accession No.	Size (bp)
*T. granosa*	T_gra_F	GTC CAC CTA ATT TGA CTG TG	Cytb	MW222474.1	105
	T_gra_R	ACG GCA TGT AAT GAC AC			
*A. broughtonii*	A_bro_F	GAG TTT TTG GTC TTG CAG GGA TA	COI	AB729113.1	112
	A_bro_R	GCA CGT CTA TCA ACA TCC ATA CCT			
*A. kagoshimensis*	A_kago_F	GAG CAC GAT AAT CAC AGC	COI	KF750628.1	109
	A_kago_R	CAC AGG CAA AGA AAG CAA C			

**Table 2 foods-11-02449-t002:** In silico specificity for three species-specific primer pairs using in silico PCR program.

Species (Accession Number)	In Silico Specificity with Primer (Amplicon Size) ^1^		
	GA	BRA	HCA
*Tegillarca granosa* (NC_026081.1)	+ (105 bp)	−	−
*Tegillarca granosa* (KJ607173.1)	+ (105 bp)	−	−
*Tegillarca granosa* (MW222474.1)	+ (105 bp)	−	−
*Anadara broughtonii* (OM807134.1)	−	+ (112 bp)	−
*Anadara broughtonii* (OM807133.1)	−	+ (112 bp)	−
*Anadara broughtonii* (OM807132.1)	−	+ (112 bp)	−
*Anadara broughtonii* (OM807131.1)	−	+ (112 bp)	−
*Anadara kagoshimensis* (OM807136.1)	−	−	+ (109 bp)
*Anadara kagoshimensis* (MN366013.1)	−	−	+ (109 bp)
*Anadara kagoshimensis* (KF750628.1)	−	−	+ (109 bp)
*Anadara antiquata* (MK783262.1)	−	−	−
*Anadara consociata* (MH535977.1)	−	−	−
*Anadara cornea* (OM807135.1)	−	−	−
*Anadara crebricostata* (MN316632.1)	−	−	−
*Anadara globosa* (MN366011.1)	−	−	−
*Anadara gubernaculum* (MN061840.1)	−	−	−
*Anadara inaequivalvis* (MN366012.1)	−	−	−
*Anadara pilula* (KU975162.1)	−	−	−
*Anadara transversa* (MN326817.1)	−	−	−
*Anadara vellicata* (KP954700.1)	−	−	−
*Arca zebra* (MN366003.1)	−	−	−
*Arctica islandica* (KF363952.1)	−	−	−
*Arctica islandica* (KF363951.1)	−	−	−
*Atrina pectinata* (KC153059.1)	−	−	−
*Barbatia decussata* (MW629559.1)	−	−	−
*Corbicula fluminea* (MK392334.1)	−	−	−
*Crassostrea gigas* (MZ497416.1)	−	−	−
*Crassostrea nippona* (HM015198.1)	−	−	−
*Dreissena polymorpha* (CM035931.1)	−	−	−
*Haliotis discus hannai* (KU310896.1)	−	−	−
*Haliotis discus hannai* (KF724723.1)	−	−	−
*Haliotis discus hannai* (KU310897.1)	−	−	−
*Mactra quadrangularis* (MN317133.1)	−	−	−
*Mactra quadrangularis* (MW691169.1)	−	−	−
*Meretrix lusoria* (MT418596.1)	−	−	−
*Mizuhopecten yessoensis* (FJ595959.1)	−	−	−
*Mytilus coruscus* (KJ577549.1)	−	−	−
*Mytilus edulis* (AY484747.1)	−	−	−
*Solen strictus* (NC_017616.1)	−	−	−
*Turbo cornutus* (MZ826276.1)	−	−	−
*Venerupis philippinarum* (AB065375.1)	−	−	−
*Venerupis philippinarum* (AB065374.1)	−	−	−

^1^ GA, granular ark; BRA, broughton’s ribbed ark; HCA, half-crenate ark. +, amplified by the corresponding primer sets; −, not amplified.

**Table 3 foods-11-02449-t003:** Specificity results of ark shells specific ultrafast PCR assay.

Common Name (Scientific Name)	Ct Value of Ultrafast PCR with Species-Specific Primer Pairs (Tm, °C) ^1^		
	Granular Ark	Broughton’s Ribbed Ark	Half-Crenate Ark
Granular ark (*T*. *granosa*)	20.41 ± 0.08 (75.96 ± 0.41)	ND ^2^	ND
Broughton’s ribbed ark (*A*. *broughtonii*)	ND	19.97 ± 0.05 (76.25 ± 0.19)	ND
Half-crenate ark (*A*. *kagoshimensis*)	ND	ND	21.53 ± 0.32 (77.72 ± 0.41)
Venus mactra (*M*. *quadrangularis*)	ND	ND	ND
Japanese littleneck (*V*. *philippinarum*)	ND	ND	ND
Hard shelled mussel (*M*. *coruscus*)	ND	ND	ND
Japanese abalone (*H*. *discus hannai*)	ND	ND	ND
Horned turban (*T*. *cornutus*)	ND	ND	ND
Ocean quahog (*A*. *islandica*)	ND	ND	ND
Lamellated oyster (*O*. *denselamellosa*)	ND	ND	ND
Kuruma prawn (*M*. *japonicus*)	ND	ND	ND
Whiteleg shrimp (*L*. *vannamei*)	ND	ND	ND
Fleshy prawn (*F*. *chinensis*)	ND	ND	ND
Gazami crab (*P*. *trituberculatus*)	ND	ND	ND
American lobster (*H*. *americanus*)	ND	ND	ND
Japanese common squid (*T*. *pacificus*)	ND	ND	ND
Pacific chub mackerel (*S*. *japonicus*)	ND	ND	ND

^1^ Mean Ct value ± standard deviation obtained from triplicate reactions. ^2^ ND, not detected.

**Table 4 foods-11-02449-t004:** Detectability of ultrafast PCR with various target species concentrations.

Common Name (Scientific Name)	DNA Concentration (ng) Extracted from Target ^1^					
	10	1	0.1	0.01	0.001	0.0001
Granular ark (*T*. *granosa*)	+++	+++	+++	+++	+++	−−−
Broughton’s ribbed ark (*A*. *broughtonii*)	+++	+++	+++	+++	+++	−−−
Half-crenate ark (*A*. *kagoshimensis*)	+++	+++	+++	+++	+++	−−−

^1^ +, amplified by the corresponding primer sets; −, not amplified.

**Table 5 foods-11-02449-t005:** Application of the direct ultrafast PCR method for identification of ark shell species in commercially processed food.

Product Type	Declared Species	Extracted DNA ^1^			Direct Method		
		GA	BRA	HCA	GA	BRA	HCA
Auto	Granular ark	+ ^2^	−	−	+	−	−
Auto	Broughton’s ribbed ark	−	+	−	−	+	−
Boiled	Granular ark	+	−	−	+	−	−
Boiled	Broughton’s ribbed ark	−	+	−	−	+	−
Boiled	Broughton’s ribbed ark	−	+	−	−	+	−
Boiled	Half-crenate ark	−	−	+	−	−	+
Boiled	Half-crenate ark	−	−	+	−	−	+
Boiled	Ark shell	−	−	+	−	−	+
Canned	Half-crenate ark	−	−	+	−	−	+
Canned	Ark shell	−	−	+	−	−	+
Dried	Granular ark	+	−	−	+	−	−
Dried	Broughton’s ribbed ark	−	+	−	−	+	−
Fried	Granular ark	+	−	−	+	−	−
Pickled	Broughton’s ribbed ark	−	+	−	−	+	−
Pickled	Half-crenate ark	−	−	+	−	−	+
Pickled	Half-crenate ark	−	−	+	−	−	+
Porridge	Ark shell	−	−	+	−	−	+
Raw	Granular ark	+	−	−	+	−	−
Raw	Granular ark	+	−	−	+	−	−
Raw	Granular ark	+	−	−	+	−	−
Raw	Broughton’s ribbed ark	−	+	−	−	+	−
Raw	Broughton’s ribbed ark	−	+	−	−	+	−
Raw	Half-crenate ark	−	−	+	−	−	+
Raw	Half-crenate ark	−	−	+	−	−	+
Seasoned	Broughton’s ribbed ark	−	+	−	−	+	−
Seasoned	Half-crenate ark	−	−	+	−	−	+
Seasoned	Half-crenate ark	−	−	+	−	−	+
Seasoned	Half-crenate ark	−	−	+	−	−	+
Seasoned	Ark shell	−	−	+	−	−	+

^1^ GA, granular ark; BRA, broughton’s ribbed ark; HCA, half-crenate ark. ^2^ +, amplified by the corresponding primer sets; −, not amplified.

## Data Availability

The data presented in this study are available on request from the corresponding author.

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
