# Peer review of "Rapid On-Site Identification for Three Arcidae Species (Anadara kagoshimensis, Tegillarca granosa, and Anadara broughtonii) Using Ultrafast PCR Combined with Direct DNA Extraction"

_foods, 2022, doi:10.3390/foods11162449_

Round 1

Reviewer 1 Report

Very interesting "on field" work.

I have 4 comments.

1. Why did you choose a mitochondrial gene for species identification? I consider a nuclear gene more appropriate for this reason. Please comment.

2. What about heteroplasmy? Did you detect signs of heteroplasmy?

3. Can this method be used for quantification of each species? Please comment.

4. Could different conservation methods contain pcr inhibitors? Please comment.

Thanks in advance.

Author Response

Response to Reviewer 1 Comments

Very interesting "on field" work.

I have 4 comments.

  1. Why did you choose a mitochondrial gene for species identification? I consider a nuclear gene more appropriate for this reason. Please comment.

Response: Mitochondrial gene was widely used as a genetic marker to detect and distinguish eukaryote species by PCR-based methods (Li et al., 2019; Wilwet et al., 2021; Kang et al., 2019). This gene is highly conserved between same species, but has a high-DNA variation in the different species. Also, mitochondrial gene can sensitively detect a target even when the DNA in the samples is degraded or mixed with a small amount of DNA in food due to the high copy number of DNA in the cell (Girish et al, 2004). As you recommended, we added the sentence in lines 169-171 as follows:

Lines 169-171: Also, the mitochondrial gene can sensitively detect a target even when the DNA in the samples is degraded or mixed with a small amount of DNA in food due to the high copy number of DNA in the cell [28].

  1. What about heteroplasmy? Did you detect signs of heteroplasmy?

Response: We designed three primers from the coding region in the mitochondrial genome. Previous studies have reported that heteroplasmy is rarely found in coding regions and is found in regions rich in A-T (Magnacca et al., 2010; Nardi et al., 2001). Mitochondrial genes contain well conserved regions and have very low interspecies variability, including well-conserved regions (Mohamad et al., 2013). Also, mitochondrial genome is stable due to it’s circular structure (Marshall et al., 2021). In silico PCR and ultrafast PCR results of our study, it seems that heterogeneity does not occur in the region as it is amplified with the same target size for different specimens.

  1. Can this method be used for quantification of each species? Please comment.

Response: The method cannot be used for the quantification of each species. The aim of this study is to develop a method that can accurately distinguish and identify three closely related ark shell species rather than quantitative analysis. As you recommended, we added the sentence in lines 317-318 as follows:

Lines 317-318: This assay developed in this study can be utilized for quantitative analysis using real-time PCR or droplet digital PCR equipment in the future.

  1. Could different conservation methods contain pcr inhibitors? Please comment.

Response: Processed foods with spice products reportedly contain inhibitors, such as carbohydrates, polysaccharides, lipids, proteins, and phenolic substances that can negatively affect the PCR (Focke et al., 2011; Kang et al., 2018). PCR inhibitors contained in processed foods may affect PCR efficiency. However, in our study, PCR inhibitors did not affect PCR efficiency, as the results of ultrafast PCR (using DNA extracted with commercial kit) and direct ultrafast PCR (without DNA extraction) are the same. The presence or absence of PCR inhibitor can be accurately possible with absolute quantification methods such as droplet digital PCR, but cannot be identified by this method. As you recommended, we added the sentence in lines 280-285 as follows:

Lines 280-285: Processed foods with spice products reportedly contain inhibitors, such as polysaccharides, carbohydrates, lipids, and proteins that can negatively affect the PCR [37,38]. PCR inhibitors contained in processed foods may affect PCR efficiency. However, in this study, ultrafast and direct ultrafast PCR efficiencies were the same, indicating that the PCR inhibitor in processed food did not affect the direct ultrafast PCR efficiency.

Reviewer 2 Report

Lines 12-13: Authors may explain better what they mean in this sentence

Lines 45-47: Authors may explain better what they mean in this sentence

Line 135: "applied" is not the appropriate verb here

Line 149: "show" instead of "showed"

Line 159: "characteristics" instead of "characteristic"

Line 195: please be specific (maybe in the Material & methods section) regarding which tool has been used

Lines 227-233: It is not clear how the standard curves have been constructed, please describe in the Materials & methods section

Lines 237-239: Authors may have used the serial dilutions to construct also the standard curves. The previous paragraph and this particular sentence may need to be rephrased. 

Line 249: change to "lower detection limit" or "higher sensitivity"

Line 269: 29 instead of 25

Line 288: "the identification of economically motivated food adulteration" instead of "economically motivated adulteration food identification"

Author Response

Response to Reviewer 1 Comments

Lines 12-13: Authors may explain better what they mean in this sentence

Response: As you recommended, we revised the sentence in lines 12-13 as follows:

Lines 12-13: granular ark exhibiting a higher economic value due to its rarity.

Lines 45-47: Authors may explain better what they mean in this sentence

Response: As you recommended, we revised the sentence in lines 42-43 as follows:

Lines 42-43: In general, fresh ark shells are consumed as processed food, steamed, boiled, or seasoned to extend the shelf-life and ensure safety.

Line 135: "applied" is not the appropriate verb here

Response: As you recommended, we revised the sentence in line 141 as follows:

Line 141: A total of 29 commercial ark shell products were used to verify the reliability of the direct ultrafast PCR assay.

Line 149: "show" instead of "showed"

Response: As you recommended, we revised the "show" instead of "showed" in line 155 as follows:

Line 155: The three species show morphological differences

Line 159: "characteristics" instead of "characteristic"

Response: As you recommended, we revised "characteristics" instead of " characteristic " in line 311 as follows:

Line 311: Morphological characteristics of granular ark

Line 195: please be specific (maybe in the Material & methods section) regarding which tool has been used

Response: As you recommended, we added the tools used in lines 197-199 as follows:

Lines 197-199: The web-based in silico PCR amplification tool (http://insilico.ehu.es/PCR/) allows confirming the specificity of the designed primer set by combining local and global alignment algorithms.

Lines 227-233: It is not clear how the standard curves have been constructed, please describe in the Materials & methods section

Response: As you recommended, we added sentences on how to construct the standard curve in the Materials and methods section.

Lines 123-127: The standard curve was generated to determine ultrafast PCR efficiency and sensitivity. The standard curve was constructed by using serial dilutions of the target DNA from 10 ng to 0.0001 ng. The different DNA concentrations were plotted against the corresponding cycle threshold (Ct) values. The amplification efficiency was calculated based on the formula: Efficiency = 101/slope−1 [25].

Lines 237-239: Authors may have used the serial dilutions to construct also the standard curves. The previous paragraph and this particular sentence may need to be rephrased. 

Response: As you recommended, we revised the sentence on the construction of the standard curve in lines 232-235 as follows:

Lines 232-235: Standard curves for granular ark, broughton’s ribbed ark, and half-crenate ark were constructed using serially diluted genomic DNA of each species (10 ng to 0.0001 ng) independently and in triplicates to confirm the accuracy of ultrafast PCR method.

Line 249: change to "lower detection limit" or "higher sensitivity"

Response: As you recommended, we revised the sentence in line 253 as follows:

Line 253: Compared with previous studies, ultrafast PCR yielded a similar or two orders of magnitude higher sensitivity than that of conventional PCR methods

Line 269: 29 instead of 25

Response: Although 29 products were used, the products specified in this sentence refer to 25 commercial products containing the species declared by the producer. The remaining four products are commercial products that did not indicate the species on the label. Thus, we removed “all” in lines 272 and 275-276 as follows:

Line 272: The analysis showed that 25 products contained the species declared by the producer (Table 5).

Lines 275-276: The four products that did not indicate the accurate species name on the label contained half-crenate ark (Anadara kagoshimensis).

Line 288: "the identification of economically motivated food adulteration" instead of "economically motivated adulteration food identification"

Response: As you recommended, we revised the sentence in line 291 as follows:

Line 291: the identification of economically motivated food adulteration.

Reviewer 3 Report

The current manuscript reports a rapid on-site identification for three Arcidae species (Anadara kagoshimensis, Tegillarca granosa, and Anadara broughtonii) using ultrafast PCR combined with direct DNA extraction.

In general, this is an important and interesting research, logically structured.

I have, however, a few comments or suggestions.

1.     Lines 66-72: clearly formulate the goal, and transfer the statement about the tasks solved and the results obtained to the conclusion

2.     In Table 1, you can remove the last column, it is not informative.

3.     Nothing is written about morphological characteristics in the research methods, maybe there is some kind of regulatory document by which they are determined?

4.     Line 261 says “29 commercial food products” and line 269 says “all 25 products”. "All" should be removed in this context, otherwise misleading

Author Response

Response to Reviewer 2 Comments

The current manuscript reports a rapid on-site identification for three Arcidae species (Anadara kagoshimensis, Tegillarca granosa, and Anadara broughtonii) using ultrafast PCR combined with direct DNA extraction.
In general, this is an important and interesting research, logically structured.
I have, however, a few comments or suggestions.

  1. Lines 66-72: clearly formulate the goal, and transfer the statement about the tasks solved and the results obtained to the conclusion

Response: As you recommended, we transferred the statement about the tasks solved and the results obtained to the conclusion and revised the sentence in lines 67-69 and 299-303 as follows:

Lines 67-69: The aim of this study is to develop a rapid and cost-effective ultrafast PCR assay combined with direct DNA extraction for field identification and the monitoring of the three ark shell species in processed products.

Lines 299-303: The specific primer sets were designed based on the genes encoding mitochondrial cytochrome b (Cytb) or cytochrome c oxidase I (COI) to distinguish granular ark, broughton’s ribbed ark, and half-crenate ark. We used the designed primer sets to develop the identification method using an ultrafast PCR machine with a microfluidic chip.

  1. In Table 1, you can remove the last column, it is not informative.

Response: As you recommended, we removed the last column (reference column) in Table 1.

  1. Nothing is written about morphological characteristics in the research methods, maybe there is some kind of regulatory document by which they are determined?

Response: As you recommended, we added the sentence in lines 74-77 as follows:

Lines 74-77: The morphological characteristics of granular ark, broughton’s ribbed ark, and half-crenate ark were investigated based on specimen information provided by the National Fisheries Research & Development Institute in Korea (https://www.nifs.go.kr/frcenter/species/).

  1. Line 261 says “29 commercial food products” and line 269 says “all 25 products”. "All" should be removed in this context, otherwise misleading

Response: As you recommended, we removed “All” in line 272 as follows:

Line 272: 25 products contained the species declared by the producer

Reviewer 4 Report

This manuscript reported the rapid on-site identification of three Arcidae species via PCR. This topic is interesting, and the experiments were designed well.

 1.      Introduction. Please reduce the first paragraph and extend the current development of PCR on this similar topic.

2.      Figure 1 is not necessary and can be moved to supporint information.

3.      It wrote that “As for the specificty, more sample species should be involved. The 29 commercial food products, classified into nine product types: raw, boiled, seasoned, pickled, canned, porridge, auto, dried, and fried.”. It is good, but nine product types can not prove that all potential fraud species can be distinguished. Thus, it suggests more Arcidae species and related species should be added for showing its specificity.

Author Response

Response to Reviewer 3 Comments

This manuscript reported the rapid on-site identification of three Arcidae species via PCR. This topic is interesting, and the experiments were designed well.

  1. Introduction. Please reduce the first paragraph and extend the current development of PCR on this similar topic.

Response: As you recommended, we reduced the first paragraph and extended the current development of PCR in lines 52-56 and 61-63 as follows:

Lines 52-56: PCR-based techniques, such as DNA barcoding, PCR restriction fragment length polymorphism, forensically informative nucleotide sequencing, and recombinase polymerase amplification, have been developed to detect different species [11–14]. These techniques have been used to identify commercially important seafood species, such as shrimps, cods, crabs, and oysters [8,15–17].

Lines 61-63: In fact, ultrafast PCR systems have been used to identify closely related eukaryotic species such as shrimp allergen, meat species, and mi-iuy croaker [18,21,22].

  1. Figure 1 is not necessary and can be moved to supporint information.

Response: As you recommended, we moved Figure 1 to supplementary materials and revised the sentence in lines 153 and 310-313 as follows:

Line 153: Figure S1 presents the morphological characteristics

Lines 310-313: Figure S1: Morphological characteristics of granular ark (Tegillarca granosa), broughton's ribbed ark (Anadara broughtonii), and half-crenate ark (Anadara kagoshimensis). (A) shells, (B) shells with meat, and (C) ark shell meats

  1. It wrote that “As for the specificty, more sample species should be involved. The 29 commercial food products, classified into nine product types: raw, boiled, seasoned, pickled, canned, porridge, auto, dried, and fried.”. It is good, but nine product types can not prove that all potential fraud species can be distinguished. Thus, it suggests more Arcidae species and related species should be added for showing its specificity.

Response: Since specimens for more Arcidae species were not available, we newly confirmed the specificity for more Arcidae species and related species by performing in silico PCR. To confirm specificity, we newly added 11 following species: Anadara antiquata (MK783262.1), Anadara consociata (MH535977.1), Anadara cornea (OM807135.1), Anadara crebricostata (MN316632.1), Anadara globosa (MN366011.1), Anadara gubernaculum (MN061840.1), Anadara pilula (KU975162.1), Anadara transversa (MN326817.1), Anadara vellicata (KP954700.1), Arca zebra (MN366003.1), and Barbatia decussata (MW629559.1).

Line 106: 32 mitochondrial genomes

Lines 110-115: A. antiquata (MK783262.1), A. consociata (MH535977.1), A. cornea (OM807135.1), A. crebricostata (MN316632.1), A. globosa (MN366011.1), A. gubernaculum (MN061840.1), A. inaequivalvis (MN366012.1), A. pilula (KU975162.1), A. transversa (MN326817.1), A. vellicata (KP954700.1), Arca zebra (MN366003.1), Arctica islandica (KF363952.1 and KF363951.1), Atrina pectinata (KC153059.1), Barbatia decussata (MW629559.1),

Line 202: 42 mitochondrial genomes

Table 2: We newly added rows for in silico specificity results in Table 2.

Round 2

Reviewer 4 Report

This work was revised well and can be consider to be published after language checking.